# Searching for Osmosensing Determinants in Poplar Histidine-Aspartate Kinases

**DOI:** 10.3390/ijms24076318

**Published:** 2023-03-28

**Authors:** Hanae Makhokh, Pierre Lafite, Mélanie Larcher, Frédéric Lamblin, Françoise Chefdor, Christiane Depierreux, Mirai Tanigawa, Tatsuya Maeda, Sabine Carpin, François Héricourt

**Affiliations:** 1Laboratoire de Biologie des Ligneux et des Grandes Cultures (LBLGC), Université d’Orléans, INRAE USC1328, 45067 Orléans Cedex 2, France; 2Institut de Chimie Organique et Analytique (ICOA), UMR CNRS-Université d’Orléans 7311, Université d’Orléans, BP 6759, 45067 Orléans Cedex 2, France; 3Department of Biology, Hamamatsu University School of Medicine, 1-20-1 Handayama, Higashi-ku, Shizuoka 431-3192, Japan

**Keywords:** histidine-aspartate kinase (HK), cache domain, extracellular domain (ECD), transmembrane (TM) domain, MultiStep Phosphorelay (MSP), Osmosensing, *Populus*, drought signaling

## Abstract

Previous works have shown the existence of protein partnership, belonging to a MultiStep Phosphorelay (MSP), potentially involved in osmosensing in *Populus*. The first actor of this signalling pathway belongs to the histidine-aspartate kinase (HK) family, which also includes the yeast osmosensor Sln1, as well as the Arabidopsis putative osmosensor AHK1. In poplar, the homologous AHK1 protein corresponds to a pair of paralogous proteins, HK1a and HK1b, exhibiting an extracellular domain (ECD), as in Sln1 and AHK1. An ECD alignment of AHK1-like proteins, from different plant species, showed a particularly well conserved ECD and revealed the presence of a cache domain. This level of conservation suggested a functional role of this domain in osmosensing. Thus, we tested this possibility by modelling assisted mutational analysis of the cache domain of the *Populus* HK1 proteins. The mutants were assessed for their ability to respond to different osmotic stress and the results point to an involvement of this domain in HK1 functionality. Furthermore, since HK1b was shown to respond better to stress than HK1a, these two receptors constituted a good system to search for osmosensing determinants responsible for this difference in efficiency. With domain swapping experiments, we finally demonstrated that the cache domain, as well as the second transmembrane domain, are involved in the osmosensing efficiency of these receptors.

## 1. Introduction

Water deficit is one of the most important environmental stresses affecting plant growth. Therefore, understanding the mechanism of drought stress perception is a major challenge to select new drought-tolerant plants. At the cellular level, this abiotic stress is perceived as an osmotic stress, and the plant mechanism of osmosensing is still unresolved, thus constituting an ongoing research field of particular interest [1,2]. So far, the best-known eukaryote osmosensing pathway corresponds to the one of the yeast *Saccharomyces cerevisiae* [3]. This pathway involves a MultiStep Phosphorelay (MSP) composed of (i) Sln1, a hybrid-type histidine-aspartate kinase (HK) osmosensor, (ii) Ypd1, a histidine-containing phosphotransfer (HPt) protein and (iii) Ssk1, the response regulator (RR). Under isotonic conditions, Sln1 is active and autophosphorylates on histidine-conserved residue in its own transmitter domain (TD), then transfers the phosphate group to its own C-terminal receiver domain (RD) [4,5]. This phosphate group is then transferred to Ypd1 and, finally, to Ssk1, leading to the inactivation of the HOG pathway. On the other hand, under hyper-osmotic stress conditions, Sln1 phosphorylation is repressed, leading to the activation of the HOG Mitogen-Activated Protein (MAP) kinase cascade [6].

In yeast, Sln1 is integrated into the membrane by two transmembrane domains, flanking an extracellular domain (ECD) in the N-terminal part [4]. It has been shown that the integrity of this ECD is essential for its sensor function, in response to change in turgor pressure [7]. Furthermore, another HK protein, AHK1, has been proposed to be involved in osmosensing in Arabidopsis since this HK is able to perceive osmotic stress in yeast, triggering HOG pathway activation [8,9], and is located at the plant plasma membrane [10]. *In planta* analysis of *ahk1* mutants gave evidence that AHK1 (i) acts as a positive regulator of osmotic stress signalling, (ii) is involved in drought tolerance [9,11] and (iii) controls plant water status, probably by determining stomatal density [12].

In *Populus*, two proteins homologous to AHK1 have been identified, namely HK1a and HK1b [13,14]. Unlike Arabidopsis, the full MSP for both poplar HK1s has been determined with HPt and RR partners [15,16,17,18,19,20]. These HK1s’ partners have been shown to be shared with the cytokinin (CK) MSP, showing interconnection between these two signalling pathways [21].

Recently, a large computational analysis on sensor receptors was performed and revealed that the so-called “extracellular PAS” (Per/Arnt/Sim) domain is indeed a cache domain [22]. This study revealed that cache domains are widely present in all kinds of living organisms and comprise the largest superfamily of extracellular sensors in prokaryotes. A cache domain was initially identified as a signalling domain able to bind small molecules, common to animal Ca^2+^ (Ca) channel subunits and prokaryotic chemotaxis (che) receptors [23]. This domain was identified in a wide range of extracellular domains of ligand-binding proteins. A secondary structure consensus has been characterized with three ββ sheets, followed by an α helix and two more β sheets. In 2016, Upadhyay and colleagues refined the existing models and established eight new classes of cache domain models: four single and four double. This new design allowed the identification of more than 50,000 new proteins bearing a cache domain. Among them, the yeast osmosensor Sln1, as well as the putative plant osmosensors from Arabidopsis (AHK1) and poplar (HK1a), were identified (HK1b was not available in the database at this time). All these data questions on the possible involvement of the cache domain in these plant receptors, and a deeper insight into this domain function, seem to be of particular interest.

Based on these findings, we performed an alignment of the ECD of AHK1 homologous proteins from different plant species, revealing a well-conserved cache domain in sequence and structure. Since AHK1 and HK1a/b were characterized as osmosensors in yeast, this alignment result allowed us to hypothesize that this cache domain may be responsible for the osmosensing capacity of these receptors. We tested this possibility by a genetic approach with a mutational analysis of poplar HK1a and HK1b cache domains. A rational mutation design, assisted by modelling, was used to direct point mutations, and mutants were tested in a complementation assay, with an osmodeficient yeast strain challenged on different osmotic stress and strength. These assays allowed us to identify positions that could be involved in protein functionality. Moreover, according to previous results demonstrating that HK1b is able to respond to osmotic stress more efficiently than HK1a [14], we performed domain swapping experiments between the two HK1s to identify the determinant of osmosensing efficiency in these two HK receptors. Furthermore, according to literature, a convergent bundle of evidence pinpoints the importance of the last transmembrane (TM) domain in HK functionality for yeast Sln1 [7], as well as plant CK receptors AHK4 (also named CRE1) [24] or AHK2 [25]. These data lead us to test the impact of such a domain in the efficiency of our poplar HK1 receptors. Thus, we determined that cache domain in the ECD, as well as in second TM domain, play a role in osmosensing functionality. Finally, the involvement of these osmosensing determinants is discussed.

## 2. Results

### 2.1. HK1a/b Contain a Conserved Cache Domain in Their ECD

According to Upadhyay and colleagues (2016), a cache domain is present in the osmosensor Sln1 and the putative plant osmosensors AHK1 and HK1a (Figure 1A). This cache domain corresponds to the model type dcache_1 (first type of double cache domain) and presents two cache sub-domains (residues 154–364 in HK1a and 161–371 in HK1b) [22]. If this domain constitutes a specific motif involved in the functionality of the receptor, one may expect that it should be present and conserved among different plant species. Consequently, we retrieved the sequence of 27 AHK1 homologous proteins from 21 plant species and performed an alignment of their respective ECD sequence, based on the sequence between the two transmembrane domains of HK1a/b. Figure 1B presents a zoom of this alignment (full ECD alignment in Appendix A), corresponding to the second cache sub-domain, and shows that ECD sequences are well-conserved across a wide range of plant species. Moreover, the conserved sequences are well superimposed with the specific secondary structures of a cache domain (β1, β2, β3, α1, β4, β5) [23], indicating that a cache domain is indeed present in the ECD of numerous HK1-like proteins. This level of conservation pinpoints the possible importance of this domain in the functionality of these receptors and could correspond to a functional osmosensing domain in plants.

### 2.2. Modelling-Assisted Rational Design of Cache Domain Mutants

Since a cache domain corresponds to small ligand-binding domain, we were interested to know whether the cache domain of our poplar HK1a/b could also bind small molecules, unknown so far, to trigger their osmosensing activity. As a first step, we tried to model the 3D structure of the full HK1a and HK1b ECD in order to define potential positions in interaction with a ligand within the cache domain. These 3D models were built using a method based on deep learning neural networks [26].

These models were then submitted to the DALI server to identify structural homologues among a subset of representative Protein Data Bank (PDB) structures (database PDB25) [27]. The closest structure to both HK1a and HK1b was the cache domain of the histamine receptor TlpQ, from *Pseudomonas aeruginosa* PAO1, with a bound histamine (PDB 6FU4) [28]. Both models were superimposed to this structure and three zones in HK1a and HK1b models in close proximity of the ligand were defined (Figure 2). Thus, we selected identical positions in both proteins, corresponding to polar or charged amino acids, that could interact with a bound ligand through electrostatic or H-bond interactions, and mutated them to alanine to suppress these. The targeted residues were K296, S304 and T324 (resp. K303, Q311, T331) in HK1a (resp. HK1b) sequences.

### 2.3. Functional Test of Cache Domain Mutants

In order to test these mutants, we performed a complementation assay in a *S. cerevisiae* deletion mutant strain (MH179) where the two proteins involved in osmosensing were deleted (*sln1*Δ *sho1*Δ). These deletions are lethal due to constitutive HOG pathway activation, but viability can be restored by expression of the phosphatase Ptp2, induced by galactose, which dephosphorylates Hog1 in the nucleus, thus inhibiting its transactivation of response genes.

This strain was transformed with plasmids carrying the wild type (WT) or mutated cDNAs of HK1a (T324A/S304A/K296A and triple mutant TSK) and HK1b (T331A/Q311A/K303A and triple mutant TQK), along with plasmids corresponding to positive (SLN1) and negative (pYX212) control. The complemented mutant yeasts were tested for their capacity to grow on normal or high-osmolarity mediums. The survival of all transformants is allowed on galactose medium (−U + Gal). On glucose medium (−U + Glu), inhibition of HOG response by Ppt2 is released and lethal phenotype can be observed, unless the expressed osmosensor restores the yeast osmodeficiency. We applied osmotic stress with increasing concentration of NaCl, as well as polyethylene glycol 6000 (PEG_6000_) infused into the medium (Figure 3).

Under NaCl stress conditions, the transformed yeast cells expressing single point mutants of the HK1a cache domain grew better than WT HK1a yeast cells, as observed at 0.6 and 0.9 M NaCl (Figure 3A). This positive effect is more obvious for triple point mutants (3M TSK). However, in the case of HK1b, the improved growth observed for HK1a mutants was not seen, even on 0.9 M NaCl medium with the triple mutant (3M TQK, Figure 3B).

To consolidate these results, we tested the response of these HK1a/b mutants with another type of osmotic stress, PEG_6000_ at 50% (*m*/*v*), infused into the medium (PEG50). This osmotic agent, as a non-permeant molecule, allows osmotic stress without the ionic component of salt stress. The osmotic strength of each medium was measured to compare the PEG stress medium to the NaCl stress medium (Appendix A). This measurement showed that the PEG50 medium osmolarity is between the NaCl 0.3 M and 0.6 M medium osmolarity. As shown in Figure 3, in the presence of PEG50, the mutations also appeared to improve osmosensing for HK1a, and a positive effect was observed even for HK1b, more notably for triple mutants. These results indicated that these ECD mutants improved the osmosensing response of HK1a and HK1b. Altogether, our results supported the involvement of the mutated positions in the response to osmotic stress.

As previously observed [14], HK1a and HK1b have divergent functionality in response to osmotic stress, with HK1b allowing a better response than HK1a. In order to determine whether the cache domain could be responsible for this difference, we tested these two receptors with exchanged cache domains by domain swapping. Therefore, HK1a with HK1b cache domain, and vice versa, were constructed and tested in our osmodeficient yeast strain (Figure 4).

Considering HK1a cache domain exchange mutant under salt stress, the growth was improved compared to WT HK1a, as observed at 0.6 M NaCl, which corresponds to the highest concentration allowing HK1a growth. The same positive effect of cache domain exchange was observed in the PEG stressed condition (Figure 4A). For the HK1b exchange mutant, the expected opposite effect was not observed, since there was no difference in growth between WT and ECD exchange mutant, even on the PEG stressed medium (Figure 4B). This could be due to the strong response of HK1b to stress, even on 0.9 M NaCl medium, which impaired the observation of any differences. Nevertheless, these results indicated that the cache domain contains important elements for the osmosensing function of HK1a/b.

### 2.4. Functional Test of TM Mutants

In order to test the impact of the TM domain in the functionality of our poplar HK1 receptors, we performed exchange domain experiments. In both HK1s, TM2 domain sequences (HK1a residues 453–475 and HK1b residues 461–483) differed from only one amino acid. Consequently, we performed the exchange of TM2 domain between HK1a and HK1b by a single point mutation (F470C mutation in HK1a and C478F mutation in HK1b). This domain swapping was also tested in combination with cache domain exchange (Figure 5).

In the case of HK1a, the TM domain exchange mutant (TM1b) exhibited better growth compared to WT, even on 0.9 M NaCl medium, overpassing cache domain exchange mutant (cache1b), which allowed a better growth on 0.6 NaCl medium only. Furthermore, this positive effect was still present on PEG50 medium (Figure 5A). However, no additive effect was observed for the double exchange mutant (cache/TM1b) on all stressed media. For HK1b, the exchange of the TM domain did not appear to decrease osmosensing, either in the presence of NaCl or PEG stresses (Figure 5B). The same reason as the one mentioned in the paragraph above could explain this fact. Thus, all these observations point to the implication that the TM2 domain has better efficiency of HK1b in response to osmotic stress.

Since no effect of the cache and TM domain exchanges are observed for HK1b, we assume that this discrepancy with HK1a exchange mutants may be due to the strong growth of HK1b, even in a stress condition as high as 0.9 M NaCl medium. Thus, in order to observe a possible effect, we challenged these HK1b mutants with stronger stress conditions (Figure 6).

With these high NaCl concentrations, we were able to observe a decreasing effect of TM domain exchange (TM1a) on HK1b. The negative effect was slightly visible on 1.2M NaCl but became more noticeable on 1.5 M NaCl medium (Figure 6). The double domain exchange mutant (cache/TM1a) did not show any difference in growth compared to WT HK1b, suggesting that the major effect observed is due to the TM2 domain, as observed in Figure 5 with HK1a. These results confirmed our assumption about the absence of effect for HK1b mutants on low NaCl concentration and strengthened our conclusion that the cache domain, and most of the TM2 domain, is involved in the osmosensing function of both HK1s.

## 3. Discussion

As sessile organisms, plants must cope with abiotic stress, such as soil salinity, drought and extreme temperatures. Drought stress is one of the most serious abiotic stresses that have profound effects on plant growth and survival. Understanding the biochemical and molecular mechanisms for drought stress perception, transduction and tolerance is still a major challenge in sustaining crop yield [2]. Faced with the osmotic stress generated by a water deficit, plants have several osmoprotection systems at their disposal, which include avoidance and tolerance to adapt to adverse conditions [29,30]. The HK1 MSP pathway could be one of the signalling pathways involved in osmotic stress perception, leading to these osmoprotection systems.

In the yeast *Saccharomyces cerevisiae*, the osmosensing pathway employs Sln1 as an osmosensor HK that monitors changes in turgor pressures [7]. Furthermore, it has been reported that cache domain, which corresponds to a small ligand-binding domain [23], is present in the extracellular region of this yeast osmosensor, as well as in plant HKs AHK1 and HK1a/b [22]. On the other hand, several studies have demonstrated the importance of TM domains in the function of HK receptors. It has been shown, by domain swapping analysis between Sln1 and the Arabidopsis CK receptor AHK4, that TM domains are essential for Sln1 function [7]. After using a bacterial expression system, as well as a yeast system, constitutively active mutants (gain-of-function) of AHK4 were isolated with mutation in the second TM domain [24]. In the same way, a forward genetic approach in Arabidopsis plants revealed novel gain-of-function mutants mutated in the last TM domain (TM4) in AHK2 [25]. Indeed, the last TM domain has a strategic position since it bridges the extracellular part to the intracellular part of the receptor, acting as a transmitter segment of the signal from the outside to the inside of the cell. In prokaryotes, the chemotaxis receptor Tlp3 from *Campylobacter jejuni* possesses a cache domain, and structural studies have revealed a downward displacement of the TM helix 2 towards the cytoplasm upon ligand binding to its cache domain [31]. This movement suggests a piston mechanism to transmit the signal through the TM domain, emphasizing the peculiar role of this domain.

By referring to these bibliographic data, in this present study, we demonstrated, experimentally, the importance of cache and TM domains in HK1 receptors of *Populus* and identified determinants that could be involved in HK1s response to osmotic stress. Through a genetic analysis, we tested point mutations in the cache domain of both HK1s and domain exchange mutants by cache and/or TM2 domains swapping. These mutants were tested in our osmodeficient yeast strain in order to carry out functional complementation assays in the presence of NaCl and PEG_6000_ stresses.

Our results showed that, in the presence of stress NaCl or PEG_6000_, the three mutated positions in HK1a and HK1b conferred an improved osmotic stress response compared to WT, supporting the implication of these amino acids in osmosensing function. These three positions correspond to homologous positions of three amino acids involved in ligand binding of the histamine chemoreceptor of *Pseudomonas aeruginosa* PAO1. Therefore, our results suggest that the amino acids in both HK1s may also be involved in the binding of a ligand, though this remains to be determined. Consistently, it has been demonstrated in *S. cerevisiae* that the cell wall protein Ccw12 can affect the activation of Sln1, allowing the authors to postulate that physical contact between Sln1 and some cell wall or periplasmic component may occur [32]. Thus, it is tempting to speculate that poplar HK1s may contact small molecules from the plant cell wall, or the apoplasm compartment, through the three amino acids identified in this study. Whether this contact is triggered or lost by stress remains to be determined. Nonetheless, whatever the perception mechanism is, our results show that elements in the cache domain, within the ECD, play a role in HK1s osmosensing function. It is noteworthy that, in *S. cerevisiae*, it has been shown that the integrity of the ECD is essential for Sln1 function [7]. Interestingly, a functional similarity between this yeast osmosensor and the Arabidopsis AHK4, upon CK presence, was shown. The plant HK is a CK receptor with global structural similarities, showing all HK proteins, but contains a CK-binding (CHASE) domain in its ECD [33,34]. Since AHK4 is mainly located at the endoplasmic reticulum in plants [10,35], and is mainly active in this sole subcellular compartment [36,37], the ability of this protein to act functionally as the yeast osmosensor is more likely due to structural resemblance, rather than true osmosensor function. However, CK HK receptors do have a role in response to osmotic stress. With gain-of-function and loss-of-function mutants of *AHK1*, *AHK2*, *AHK3* and *AHK4*, it was shown in Arabidopsis that AHK1 acts as a positive regulator in osmotic stress response and AHK2-3-4 act as negative regulators [9]. These antagonistic actions led to the hypothesis of a Yin-Yang balance between CK homeostasis and plant drought adaptation [38]. For our poplar HK1s, as a first step in the *in planta* involvement in drought response, it would be interesting to test these receptors in an Arabidopsis *ahk1* mutant line by complementation assay in stress conditions. In such *in planta* assays, the use of the cache domain mutants characterized in this study would allow us to confirm the contribution of these domains to the osmotic stress response in a plant background. Furthermore, the study of these two receptors, directly into *Populus* lines, would definitely bring information in the genuine homologous background. Such analyses with poplar mutant lines (overexpressed or knock-out) are currently under investigation.

In order to identify the determining factors in the differential response to osmotic stress between HK1a and HK1b, we tested mutants by cache and/or TM2 domain swapping. We showed that HK1a, with cache or TM2 domains of HK1b, exhibited a better tolerance to osmotic stress than WT HK1a. This result implies that these domains are responsible, at least in part, for the better osmosensing efficiency of HK1b. These results were in agreement with those observed for Sln1 [7], AHK4 [24] and AHK2 [25], where the last TM domain is important for HKs function. In the case of HK1b, as we assumed, the decreasing effect was observed only on higher concentrations of NaCl (1.2 or 1.5 M), revealing the importance of these two domains in HK1b function too. Finally, yeasts expressing HK1a with HK1b cache, and TM2 domains, showed a better tolerance to osmotic stresses, but still less than that observed for yeasts expressing WT HK1b. This suggests that some other elements in HK1b must be involved in its response. This observation fully supports previous results about HK1b efficiency compared to HK1a [14]. Indeed, it was demonstrated that HK1b was more permissive towards canonical phosphorylation sites than HK1a, and that other amino acids could provide alternative phosphorylation sites. Therefore, better efficiency determinants of HK1b may lie in the cytoplasmic part of this receptor.

In summary, this study provided clues about the involvement of cache domain in the functionality of HK1 receptors. We found three amino acids in this domain that may play a role in osmotic stress response. This result shows the functional importance of this domain in poplar HK1s and, given that a cache domain is present in a wide range of plant HK1-like proteins, we can hypothesize that the involvement of this domain may be a general feature in plant osmosensing. Furthermore, the TM2 domain appears to be one of the factors responsible for the differential response observed between both HK1s. These new results shed some light on the functionality of HK1 receptors in osmosensing.

As a next step, it could be interesting to explore the role of the cytoplasmic part of the HK1b receptor in the differential response, alone or in combination with other domains (cache or TM2), so we can precisely define all the determining elements in its efficiency. Moreover, further experiments at the structural level could bring valuable information on cache domain function. Structural determination of such a domain in plants would definitively provide precious insights since no structural data exist for a plant cache domain in the PDB. This approach is currently under progress. Finally, since this cache domain is involved in osmosensing and corresponds to a small ligand-binding domain, the identification of such a ligand would provide unprecedented clues to understanding the osmosensing mechanism. We are currently exploring this question with the use of an original reporter system to screen for molecules able to bind poplar HK1s. The determination of the true ligand responsible for osmotic stress response will definitely lead to valuable applications in developing drought tolerant plants for sustainable agriculture.

## 4. Materials and Methods

### 4.1. Identification of Cache Domain in Plant AHK1-like Proteins

According to Upadhyay et al. (2016), a cache domain is present in poplar HK1a from aa 154 to aa 364. Therefore, we aligned sequences of HK1a/b ECD (starting from the end of the first TM domain to the beginning of the second TM domain) with ECD sequences from 27 plant AHK1-like proteins. Amino acid sequences of HK1a (accession No AJ937747) and HK1b (accession No LT622839) were aligned by Clustal Omega (www.ebi.ac.uk/Tools/services/rest/clustalo, accessed on 13 February 2023) or Clustal W (https://npsa-prabi.ibcp.fr, accessed on 13 February 2023) with Arabidopsis (*A. thaliana*, AT2G17820), saltwater cress (*T. halophila*, XM_006409179), papaya (*C. papaya*, evm.TU.supercontig_6.277), cocoa (*T. cacao*, Thecc1EG000177), cotton (*G. raimondii*, Gorai.007G049200 and Gorai.003G008400), peach (*P. persica*, Prupe.7G170700), apple (*M. domestica*, MDP0000276711), rose gum (*E. grandis*, Eucgr.I02335), grape vine (*V. vinifera*, GSVIVG01018749001), tomato (*S. lycopersicum*, Solyc02g083680), Madagascar periwinkle (*C. roseus*, AF534893), yellow monkeyflower (*M. guttatus*, Migut.L00691), purple willow (*S. purpurea*, SapurV1A.0698s0020 and SapurV1A.0130s0330), castor bean (*R. communis*, 29656.t000014), cassava (*M. esculenta*, Manes.02G106100 and Manes.01G147600), flax (*L. usitatissimum*, Lus10041891.g and Lus10028438.g), barrel clover (*M. truncatula*, Medtr5g022470 and Medtr8g075340), common bean (*P. vulgaris*, Phvul.002G107100 and Phvul.003G264600) and soybean (*G. max*, Glyma01g36950, Glyma11g08310 and Glyma02g05220). A portion of this alignment is shown in Figure 1.

The determination of secondary structures was performed with the “Secondary structure consensus prediction” program of NPS@ web server from the Pôle Rhône-Alpes de BioInformatique (https://npsa-prabi.ibcp.fr, accessed on 13 February 2023).

### 4.2. Modelling-Assisted Rational Design of Mutants

The ECD peptide sequences of HK1a (residues 100–451) and HK1b (residues 107–458) (uniprot accession codes Q571R6 and A0A1M4NDG0, respectively) were submitted to the ROBETTA server (http://robetta.bakerlab.org), and the 3D structure prediction was based on RoseTTAFold deep learning methodology [24]. The best models, according to the Robetta score, out of the 5 generated for each protein, were selected for analysis. Briefly, Pymol software (https://www.pymol.org/pymol, accessed on 13 February 2023) was used to superimpose the HK1a and HK1b ECD models to the *Pseudomonas aeruginosa PAO1* cache domain bound with histamine. Selection of the residues, located 5 Å from the histamine in HK1a and HK1b models, enabled the identification of the putative ligand binding regions that were targeted for site-directed mutagenesis.

### 4.3. Construction of HK1a/b Mutants

Point mutations in HK1a and HK1b full-length CDS sequences were introduced by PCR with specific mutated primers (Appendix A) to create single mutants into pGEM-T (Promega, Madison, WI, USA) plasmid. The point mutations for HK1a mutants (resp. HK1b mutants) were as follows: K296A = AA886-887GC, S304A = T910G and T324A = A970G (resp. K303A = AA907-908GC, Q311A = CA931-932GC and T331A = A991G). Then, triple mutants were obtained after 3 rounds of mutations by site-directed mutagenesis, using primers carrying the mutation of interest. The resulting mutated HK1a/b constructs were subcloned into the yeast plasmid pYX212 (*Eco*RI-*Hind*III) by homologous recombination in yeast. DNA sequencing confirmed the presence of the desired mutations and that no undesired mutations were present.

Domain exchange mutations in HK1a and HK1b sequences were introduced by PCR, with specific primers (Appendix A), to create mutants by cache/TM2 domain exchange. TM mutants for HK1a: F470C = T1409G. TM mutants for HK1b: C478F = G1433T. Cache domain mutants for HK1a: cache domain HK1b residues 121–437. Cache domain mutants for HK1b: cache domain HK1a residues 121–445. To create the double mutant by domain exchange HK1b, the mutated sequence was recovered from yeast by PCR, then cloned into pJET1.2/blunt vector (Thermo Fisher Scientific, Waltham, MA, USA) by simple ligation to obtain the simple mutant by domain exchange HK1b, on which was performed a site-directed mutagenesis to create the double mutant. For HK1a double mutant, mutation was introduced on the simple mutant by PCR, with specific mutated primers. The resulting mutated sequences were cloned into the yeast plasmid pYX212 (*Eco*RI-*Hind*III) by homologous recombination. The correct mutated sequences were confirmed by sequencing.

### 4.4. Complementation Analysis of the sln1∆ sho1∆ Deletion Mutant MH179

The pPD2133 plasmid, expressing the Sln1 osmosensor, was used as a positive control (pPD-Sln1) and the empty pYX212 vector was used as a negative control. The yeast strain MH179 (*ura3 leu2 his3 sln1::LEU2 sho1::TRP1* + pGP22, with pGP22: *GALp-PTP2/pRS413 (HIS3*, *CEN)*) was used for transformation, according to [39], with all the constructs expressing WT or mutated HK1a/1b. Yeast cells were grown on galactose-containing medium lacking uracil (−U + Gal), for transformation control, and spotted onto −U + Glu medium in the absence or presence of 0.3 M, 0.6 M, 0.9 M NaCl for 4 days at 30 °C, for osmosensing complementation tests. When 1.2 or 1.5 M NaCl was applied, 5 to 7 days of culture were necessary. Tests with PEG50 medium were performed on −U + Glu medium infused with Polyethylene glycol 6000 (50% *w*/*v*) overnight. Three 10-fold dilution series, starting from an OD_600_ 0.2, was realized and 5 µL of each were spotted onto medium.

The osmolarity measurement was performed with a micro-osmometer (Roebling type 13DR) on liquid solutions. The PEG gel medium was melted before measurement. Results are the mean of 3 repeats.

## Figures and Tables

**Figure 1 ijms-24-06318-f001:**
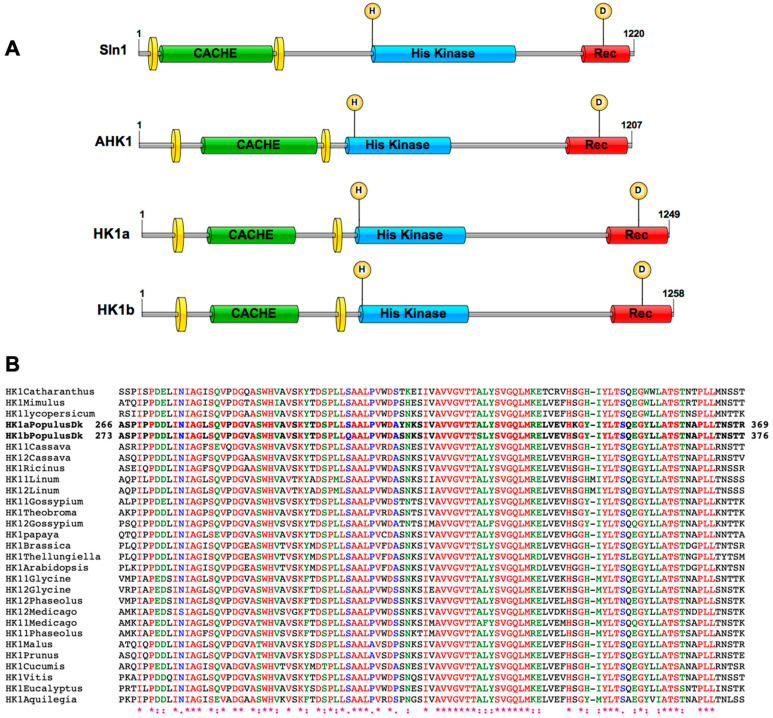
A cache domain is present in plant AHK1-like proteins. (**A**) Protein organization of Sln1 (Uniprot ID P39928), AHK1 (Uniprot ID Q9SXL4), HK1a (Uniprot ID Q571R6) and HK1b (Uniprot ID A0A1M4NDG0) is shown. All domain lengths were identified on the InterProScan analysis, except for cache domains which are based on Upadhyay et al. (2016). Transmembrane domains are represented by yellow discs; CACHE: cache domain; His Kinase: histidine kinase domain; Rec: Receiver domain. Phosphorylation sites are shown with an H (histidine) and D (Aspartate) in a yellow circle. (**B**) Alignment of HK1a/b ECD with homologous receptors in different plant species. Sequences from poplar are indicated in bold and amino acids positions of this ECD sequence segment are indicated on the left and right of these poplar sequences. Species names and sequence references are presented in the Section 4. * indicate identical amino acids, : indicate strongly similar amino acids and . indicate weakly similar amino acids. Letters in colour correspond to conserved amino acids and black letters to non-conserved amino acids.

**Figure 2 ijms-24-06318-f002:**
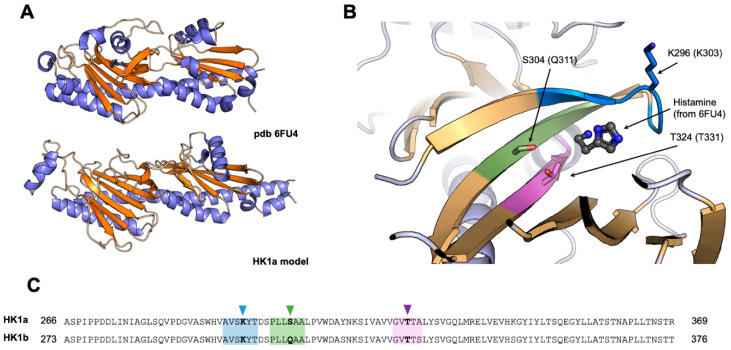
Selection of mutants involved in ligand binding in HK1a and HK1b. (**A**) Structural comparison between HK1a model and *P. aeruginosa* histamine receptor (PDB 6FU4). Helices and sheets are respectively coloured in blue and orange. (**B**) Close view of the putative binding site of HK1a, with the histamine bound from 6FU4, after the structural alignment of HK1a model and 6FU4 structure. The closest zones in contact with histamine are respectively coloured in blue, green and pink. The selected residues for mutagenesis are depicted in ball and sticks and labelled with the HK1a sequence numbering (HK1b). (**C**) Sequence alignment of HK1a and HK1b. The zones highlighted in panel (**B**) are identically coloured, and mutated residues are highlighted with arrows.

**Figure 3 ijms-24-06318-f003:**
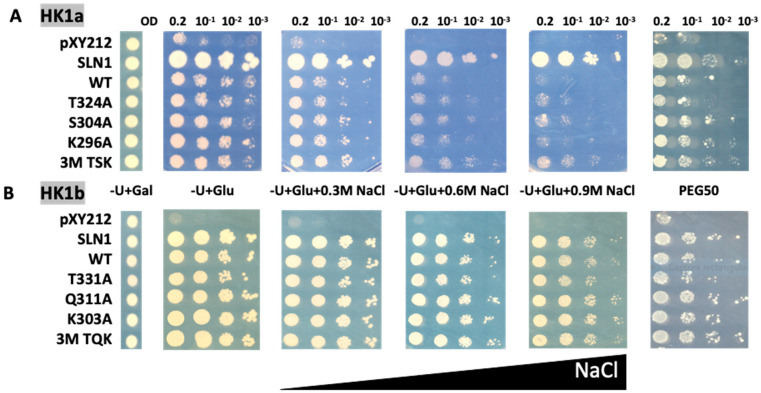
Functional complementation test for cache mutants of HK1a (**A**) and HK1b (**B**) in presence of NaCl or PEG_6000_ stresses. The osmodeficient strain (MH179) was transformed with the negative (pYX212 empty vector) and positive (SLN1) controls, as well as with wild type constructs (WT) and single point mutation constructs of HK1a (T324A, S304A, K296A) and HK1b (T331A, Q311A, K303A), or triple point mutation constructs of HK1a (3M TSK) and HK1b (3M TQK). Transformants were spotted onto galactose containing medium (−U + Gal) for growth control and onto glucose containing medium, without (−U + Glu) or with increasing NaCl concentrations (0.3, 0.6 and 0.9 M), or infused PEG_6000_ 50% *w*/*v* (PEG50), for osmotic tolerance test. Growth tests are representative of four independent replicates as serial 10-fold dilutions from an OD_600_ = 0.2, as indicated.

**Figure 4 ijms-24-06318-f004:**
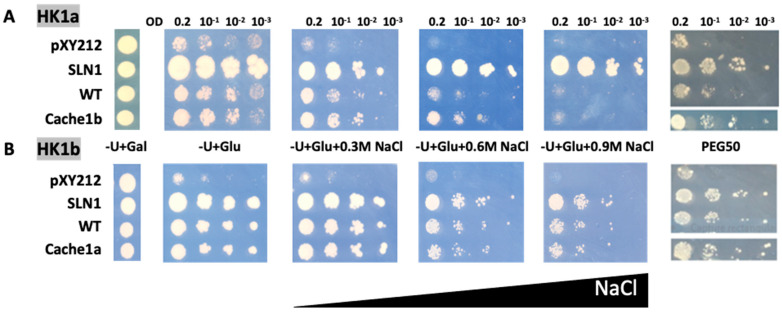
Functional complementation test for cache domain exchange mutants of HK1a (**A**) and HK1b (**B**) in presence of NaCl or PEG_6000_ stresses. The osmodeficient strain (MH179) was transformed with the negative (pYX212 empty vector) and positive (SLN1) controls, as well as with wild type constructs (WT) and cache domain exchange constructs (HK1a: cache1b residues 121–437 and HK1b: cache1a residues 121–445). Transformants were spotted onto galactose containing medium (−U + Gal) for growth control and onto glucose containing medium (−U + Glu), without or with increasing NaCl concentrations (0.3, 0.6 and 0.9 M) or infused PEG_6000_ 50% *w*/*v* (PEG50), for osmotic tolerance test. Growth tests are representative of four independent replicates as serial 10-fold dilutions from an OD_600_ = 0.2, as indicated.

**Figure 5 ijms-24-06318-f005:**
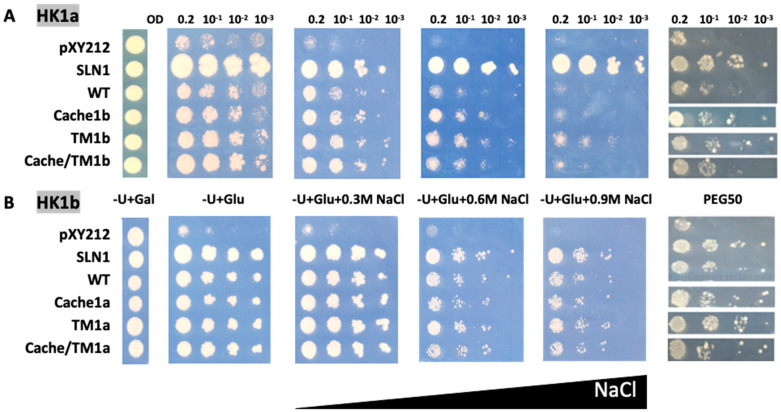
Functional complementation test for TM mutants of HK1a (**A**) and HK1b (**B**) in presence of NaCl or PEG_6000_ stresses. The osmodeficient strain (MH179) was transformed with the negative (pYX212 empty vector) and positive (SLN1) controls, as well as with wild type constructs (WT) and single cache domain exchange constructs (HK1a: cache1b residues 121–437 and HK1b: cache1a residues 121–445), TM2 exchange constructs (HK1a: TM1b with F470C mutation and HK1b: TM1a with C478F mutation) or double exchange constructs of HK1a cache/TM1b) and HK1b (cache/TM1a). Transformants were spotted onto galactose containing medium (−U + Gal) for growth control and onto glucose containing medium (−U + Glu), without or with increasing NaCl concentrations (0.3, 0.6 and 0.9 M) or infused PEG_6000_ 50% *w*/*v* (PEG50), for osmotic tolerance test. Growth tests are representative of four independent replicates as serial 10-fold dilutions from an OD_600_ = 0.2, as indicated.

**Figure 6 ijms-24-06318-f006:**
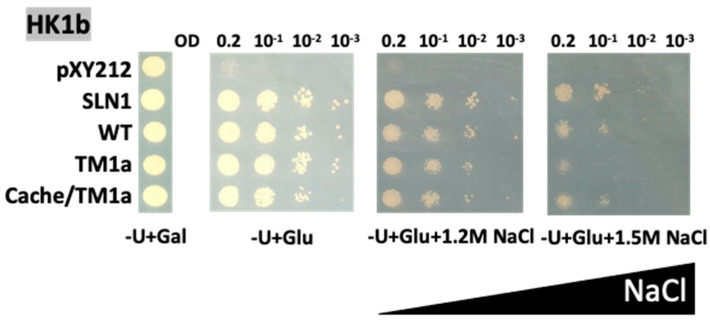
Functional complementation test for domain exchange mutants of HK1b on strong NaCl stress medium. The osmodeficient strain (MH179) was transformed with the negative (pYX212 empty vector) and positive (SLN1) controls, as well as with wild type constructs (WT) and single cache domain exchange constructs (cache1a residues 121–445), TM2 exchange constructs (TM1a with C478F mutation) or double exchange constructs of HK1b (cache/TM1a). Transformants were spotted onto galactose containing medium (−U + Gal) for growth control and onto glucose containing medium (−U + Glu), with increasing NaCl concentrations (1.2 and 1.5 M), for osmotic tolerance test. Growth tests are representative of three independent replicates as serial 10-fold dilutions from an OD_600_ = 0.2, as indicated.

## Data Availability

No new data were created or analyzed in this study. Data sharing is not applicable to this article.

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
