# Peer review of "Searching for Osmosensing Determinants in Poplar Histidine-Aspartate Kinases"

_ijms, 2023, doi:10.3390/ijms24076318_

Round 1

Reviewer 1 Report

The manuscript of Makhokh et al. presents the identification about osmosensing determinants characteristics of Populus HK1 proteins, the Cache domain with unknown function. The authors present very interesting and useful data consistent with currently available knowledge about the role of HK in yeast and Arabidopsis, particularly in the Cache domain response to different osmotic stress. Although, the work is valuable but requires a few corrections before being published.

- It is very nice to have several cache domain mutants as shown in the manuscript. The mutants were assessed for their ability to respond to different osmotic stress in yeast and the results point to an involvement of this domain in HK1 functionality. However, I suggest the author generate and include the complementation lines of Arabidopsis ahk1 mutants with Populus HK1a/b in all experiments to confirm that the Cache domain contributes to the osmotic stress response.

Minor issues:

1. Lines 100-101 –“ two Cache 100 sub-domains (residues 154-364 in HK1a and 161-371 in HK1b)”. What is the reference or supporting data for the statement?

2. In fig. 1b, Cache sub-domain sequences are well-conserved across a wide range of plant species, is there any report on what are the key amino acids with the osmosensing capacity? If have, please label it.

3. Lines 129-132, Whether the material method section is more appropriate?

4. Line 387, “Point mutations in HK1a and HK1b sequences…”. Is it their full-length CDS sequence? Please specify.

Author Response

“I suggest the author generate and include the complementation lines of Arabidopsis ahk1 mutants with Populus HK1a/b in all experiments to confirm that the Cache domain contributes to the osmotic stress response.”

We agree that complementation experiments with the use of ahk mutant lines would be of great interest to further characterize the poplar HKs. It could be a good system to confirm the contribution of Cache domain to the osmotic stress response in planta. We are also interested in testing these receptors directly in Populus as the genuine plant background. This kind of in planta experiments are currently considered for another publication. In consequence, we added a paragraph dedicated to this point in the discussion section.

Minor issues:

  1. Lines 100-101 – “two Cache sub-domains (residues 154-364 in HK1a and 161-371 in HK1b)”. What is the reference or supporting data for the statement?

This numbering corresponds to the full Cache domain, containing the two sub-domains, which was extracted from the supplemental data of the reference Upadhyay et al. 2016.

We added this reference in the text.

  1. In fig. 1b, Cache sub-domain sequences are well-conserved across a wide range of plant species, is there any report on what are the key amino acids with the osmosensing capacity? If have, please label it.

This article shows for the first time the involvement of a Cache domain in osmosensing and the identification of important amino acids in this domain, therefore there is no other reports on key amino acids with the osmosensing capacity.

  1. Lines 129-132, Whether the material method section is more appropriate?

We agree that this sentence could be more appropriate in the Mat&Met section. Accordingly, the sentence in the results section was simplified and information was reported in the Mat&Met section.

  1. Line 387, “Point mutations in HK1a and HK1b sequences…”. Is it their full-length CDS sequence? Please specify.

Yes, the mutations were introduced into the full-length CDS sequence. We added this statement in the text.

Reviewer 2 Report

This research identifies mutations of the conserved Cache domain in the Populus compared to 21 different plant sequences. And there are findings to answer the question of whether these mutations are related to osmotic sensors.

The topics studied are interesting and very few studies are found. Hk1a and HK1b receptors Cache domain were tested on yeast strain and these receptors were found related to osmosensing function characteristics. Also, the differences between Hk1a and Hk1b are due to a single point mutation which is affected by NaCl concentrations.

Although the research was prepared for the purpose, more detailed information should be given about the role of the osmosensor mechanism in plants. The discussion should be improved and the text should be written more clearly.

The manuscript gives a preview about osmosensor in plant mechanism in this state, but it would be useful to verify these regions related to osmotic stresses in the Cache domain with different methods such as qPCR.

Author Response

“Although the research was prepared for the purpose, more detailed information should be given about the role of the osmosensor mechanism in plants. The discussion should be improved and the text should be written more clearly.”

As mentioned in the introduction, details on plants osmosensing mechanism are still unclear. However, we added two references in the introduction section which correspond to reviews presenting what is known and explaining that plant osmosensing is still an open field of research.

Furthermore, the discussion has been improved by the addition of a whole paragraph dedicated to the in planta characterization of our HKs and their mutants. This paragraph was written as clearly as possible and all the modifications done in the text in response to all reviewers should render this article clearer.

“The manuscript gives a preview about osmosensor in plant mechanism in this state, but it would be useful to verify these regions related to osmotic stresses in the Cache domain with different methods such as qPCR.”

We agree that it would be useful to verify with different methods such as qPCR but we have already analyzed the poplar receptors, and therefore their Cache domain, by RT-PCR in previous articles. In Chefdor et al. (2006) and Héricourt et al. (2016), the expression of these HKs in poplar was shown in all plant organs tested and this expression was induced by osmotic stress.

Reviewer 3 Report

well focussed well written and formatted paper   thank  you 

few minor comments as sticky notes

Author Response

“Few minor comments as sticky notes.”

All comments have been answered and text has been modified accordingly.